# FedConv: Enhancing Convolutional Neural Networks for Handling Data Heterogeneity in Federated Learning

## Abstract

Federated learning (FL) is an emerging paradigm in machine learning, where a shared model is collaboratively learned using data from multiple devices to mitigate the risk of data leakage. While recent studies posit that Vision Transformer (ViT) outperforms Convolutional Neural Networks (CNNs) in addressing data heterogeneity in FL, the specific architectural components that underpin this advantage have yet to be elucidated. In this paper, we systematically investigate the impact of different architectural elements, such as activation functions and normalization layers, on the performance within heterogeneous FL. Through rigorous empirical analyses, we are able to offer the first-of-its-kind general guidance on micro-architecture design principles for heterogeneous FL. Intriguingly, our findings indicate that with strategic architectural modifications, pure CNNs can achieve a level of robustness that either matches or even exceeds that of ViTs when handling heterogeneous data clients in FL. Additionally, our approach is compatible with existing FL techniques and delivers state-of-the-art solutions across a broad spectrum of FL benchmarks.

## 1 Introduction

Federated Learning (FL) is an emerging paradigm that holds significant potential in safeguarding user data privacy in a variety of real-world applications, such as mobile edge computing (Li et al., 2020a). Yet, one of the biggest challenges in FL is data heterogeneity, making it difficult to develop a single shared model that can generalize well across all local devices. While numerous solutions have been proposed to enhance heterogeneous FL from an optimization standpoint (Li et al., 2020c; Hsu et al., 2019), the recent work by Qu et al. (2022b) highlights that the selection of neural architectures also plays a crucial role in addressing this challenge. This study delves into the comparative strengths of Vision Transformers (ViTs) vis-à-vis Convolutional Neural Networks (CNNs) within the FL context and posits that the performance disparity between ViTs and CNNs amplifies with increasing data heterogeneity.

The analysis provided in Qu et al. (2022b), while insightful, primarily operates at a macro level, leaving certain nuances of neural architectures unexplored. Specifically, the interplay between individual architectural elements in ViT and its robustness to heterogeneous data in FL remains unclear. While recent studies (Bai et al., 2021; Zhang et al., 2022) suggest that self-attention-block is the main building block that contributes significantly to ViT's robustness against out-of-distribution data, its applicability to other settings, particularly heterogeneous FL, is yet to be ascertained. This gap in understanding prompts us to consider the following questions: *which architectural elements in ViT underpin its superior performance in heterogeneous FL, and as a step further, can CNNs benefit from incorporating these architectural elements to improve their performance in this scenario?*

To this end, we take a closer analysis of the architectural elements of ViTs, and empirically uncover several key designs for improving model robustness against heterogeneous data in FL. First, we note the design of activation functions matters, concluding that smooth and near-zero-centered activation functions consistently yield substantial improvements. Second, simplifying the architecture by completely removing all normalization layers and retraining a single activation function in each block emerges as a beneficial design. Third, we reveal two key properties — feature extraction from

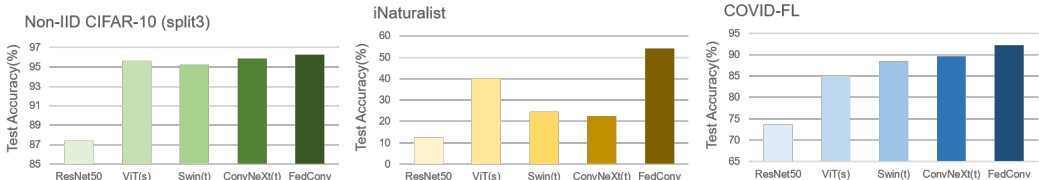

Figure 1: Performance comparison on three heterogeneous FL datasets. While a vanilla ResNet significantly underperforms Transformers and ConvNeXt when facing data heterogeneity, our enhanced CNN model, named FedConv, consistently achieves superior performance.

overlapping patches and convolutions only for downsampling — that define a robust stem layer for handling diverse input data. Lastly, we critically observe that a large enough kernel size (*i.e.*, 9) is essential in securing model robustness against heterogeneous distributions.

Our experiments extensively verify the consistent improvements brought by these architectural designs in the context of heterogeneous FL. Specifically, by integrating these designs, we are able to build a self-attention-free CNN architecture, dubbed FedConv, that outperforms established models, including ViT (Dosovitskiy et al., 2020), Swin-Transformer (Liu et al., 2021), and ConvNeXt(Liu et al., 2022). Notably, our FedConv achieves 92.21% accuracy on COVID-FL and 54.19% on iNaturalist, outperforming the next-best solutions by 2.57% and 13.89%, respectively. Moreover, our FedConv can effectively generalize to other datasets with varying numbers of data clients, from as few as 5 to more than 2,000; when combined with existing FL methods, our FedConv consistently registers new performance records across a range of FL benchmarks.

In conclusion, our findings shed light on the pivotal role of architecture configuration in robustifying CNNs for heterogeneous FL. We hope that our insights will inspire the community to further probe the significance of architectural nuances in different FL settings.

## 2 RELATED WORKS

**Federated Learning.** FL is a decentralized approach that aims to learn a shared model by aggregating updates or parameters from locally distributed training data (McMahan et al., 2017). However, one of the key challenges in FL is the presence of data heterogeneity, or varying distribution of training data across clients, which has been shown to cause weight divergence (Zhao et al., 2018) and other optimization issues (Hsieh et al., 2020) in FL.

To address this challenge, FedProx (Li et al., 2020c) adds a proximal term in the loss function to achieve more stable and accurate convergence; FedAVG-Share (Zhao et al., 2018) keeps a small globally-shared subset amongst devices; SCAFFOLD (Karimireddy et al., 2020) introduces a variable to both estimate and correct update direction of each client. Beyond these, several other techniques have been explored in heterogeneous FL, including reinforcement learning (Wang et al., 2020a), hierarchical clustering (Briggs et al., 2020), knowledge distillation (Zhu et al., 2021; Li & Wang, 2019; Qu et al., 2022a), and self-supervised learning (Yan et al., 2023; Zhang et al., 2021).

Recently, a new perspective for improving heterogeneous FL is by designing novel neural architectures. Li et al. (2021) points out that updating only non-BatchNorm (BN) layers significantly enhances FedAVG performance, while Du et al. (2022) suggests that simply replacing BN with LayerNorm (LN) can mitigate external covariate shifts and accelerate convergence. Wang et al. (2020b) shows neuron permutation matters when averaging model weights from different clients. Our research draws inspiration from Qu et al. (2022b), which shows that Transformers are inherently stronger than CNNs in handling data heterogeneity (Qu et al., 2022b). Yet, we come to a completely different conclusion: with the right architecture designs, CNNs can be comparable to, or even more robust than, Transformers in the context of heterogeneous FL.

**Vision Transformer.** Convolutional Neural Networks have been the dominant architecture in visual recognition for nearly a decade due to their superior performance (Simonyan & Zisserman, 2014; He et al., 2016; Szegedy et al., 2016; Howard et al., 2017; Tan & Le, 2019). However, the recently emerged Vision Transformers challenge the leading position of CNNs (Dosovitskiy et al., 2020; Touvron et al., 2021a;b; Liu et al., 2021) — by applying a stack of global self-attention blocks (Vaswani et al., 2017) on a sequence of image patches, Transformers can even show stronger per-

formance than CNNs in a range of visual benchmarks, especially when a huge amount of training data is available (Dosovitskiy et al., 2020; Touvron et al., 2021a; Liu et al., 2021; He et al., 2022). Additionally, Transformers are shown to be inherently more robust than CNNs against occlusions, adversarial perturbation, and out-of-distribution corruptions (Bhojanapalli et al., 2021; Bai et al., 2021; Zhang et al., 2022; Paul & Chen, 2022).

**Modernized CNN.** Recent works also reignite the discussion on whether CNNs can still be the preferred architecture for visual recognition. Wightman et al. (2021) find that by simply changing to an advanced training recipe, the classic ResNet-50 achieves a remarkable 4% improvement on ImageNet, a performance that is comparable to its DeiT counterpart (Touvron et al., 2021a). ConvMixer (Trockman & Kolter, 2022), on the other hand, integrates the patchify stem setup into CNNs, yielding competitive performance with Transformers. Furthermore, ConvNeXt (Liu et al., 2022) shows that, by aggressively incorporating every applicable architectural design from Transformer, even the pure CNN can attain excessively strong performance across a variety of visual tasks.

Our work is closely related to ConvNeXt (Liu et al., 2022). However, in contrast to their design philosophy which aims to build Transformer-like CNNs, our goal is to pinpoint a core set of architectural elements that can enhance CNNs, particularly in the context of heterogeneous FL.

## 3 FEDCONV

In this section, we conduct a comprehensive analysis of several architectural elements in heterogeneous FL. Our findings suggest that pure CNNs can achieve comparable or even superior performance in heterogeneous FL when incorporating specific architectural designs. Key designs including switching to smooth and near-zero-centered activation functions in Section 3.2, reducing activation and normalization layers in Section 3.3, adopting the stem layer setup in Section 3.4, and enlarging the kernel size in Section 3.5. By combining these designs, we develop a novel CNN architecture, **FedConv**. As demonstrated in Section 3.6, this architecture emerges as a simple yet effective alternative to handle heterogeneous data in FL.

### 3.1 EXPERIMENT SETUP

**Datasets.** Our main dataset is COVID-FL (Yan et al., 2023), a real-world medical FL dataset containing 20,055 medical images, sourced from 12 hospitals. Note that clients (*i.e.*, hospital) in this dataset are characterized by the absence of one or more classes. This absence induces pronounced data heterogeneity in FL, driven both by the limited overlap in client partitions and the imbalanced distribution of labels. To provide a comprehensive assessment of our approach, we report performance in both the centralized training setting and the distributed FL setting.

**Federated learning methods.** We consider the classic Federated Averaging (FedAVG) (McMahan et al., 2017) as the default FL method, unless specified otherwise. FedAVG operates as follows: 1) the global server model is initially sent to local clients; 2) these clients next engage in multiple rounds of local updates with their local data; 3) once updated, these models are sent back to the server, where they are averaged to create an updated global model.

**Training recipe.** All models are first pre-trained on ImageNet using the recipe provided in (Liu et al., 2022), and then get finetuned on COVID-FL using FedAVG. Specifically, in fine-tuning, we set the base learning rate to 1.75e-4 with a cosine learning rate scheduler, weight decay to 0.05, batch size to 64, and warmup epoch to 5. Following (Qu et al., 2022b; Yan et al., 2023), we apply AdamW optimizer (Loshchilov & Hutter, 2017) to each local client and maintain their momentum term locally. For FedAVG, we set the total communication round to 100, with a local training epoch of 1. Note that all 12 clients are included in every communication round.

**Computational cost.** We hereby use FLOPs as the metric to measure the model scale. To ensure a fair comparison, all models considered in this study are intentionally calibrated to align with the FLOPs scale of ViT-Small (Dosovitskiy et al., 2020), unless specified otherwise.

To improve the performance-to-cost ratio of our models, we pivot from the conventional convolution operation in ResNet, opting instead for depth-wise convolution. This method, as corroborated by prior research (Howard et al., 2017; Sandler et al., 2018; Howard et al., 2019; Tan & Le, 2019), exhibits a better balance between performance and computational cost. Additionally, following the

design philosophy of ResNeXt (Xie et al., 2017), we adjust the base channel configuration across stages, transitioning from (64, 128, 256, 512) to (96, 192, 384, 768); we further calibrate the block number to keep the total FLOPs closely align with the ViT-Small scale.

**An improved ResNet baseline.** Building upon the adoption of depth-wise convolutions, we demonstrate that further incorporating two simple architectural elements from ViT enables us to build a much stronger ResNet baseline. Firstly, we replace BN with LN. This shift is motivated by prior studies which reveal the adverse effects of maintaining the EMA of batch-averaged feature statistics in heterogeneous FL (Li et al., 2021; Du et al., 2022); instead, they advocate that batch-independent normalization techniques like LN can improve both the performance and convergence speed of the global model (Du et al., 2022). Secondly, we replace the traditional ReLU activation function with GELU (Hendrycks & Gimpel, 2016). As discussed in Section 3.2, this change can substantially increase the classification accuracy in COVID-FL by 4.52%, from 72.92% to 77.44%.

We refer to the model resulting from these changes as **ResNet-M**, which will serve as the default baseline for our subsequent experiments. However, a critical observation is that, despite its enhancements, the accuracy of ResNet-M still lags notably behind its Transformer counterpart in heterogeneous FL, which registers a much higher accuracy at 88.38%.

## 3.2 ACTIVATION FUNCTION

We hereby investigate the impact of activation function selection on model performance in the context of heterogeneous FL. Our exploration begins with GELU, the default activation function in Transformers (Dosovitskiy et al., 2020; Touvron et al., 2021a; Liu et al., 2021). Previous studies show that GELU consistently outperforms ReLU in terms of both clean accuracy and adversarial robustness (Hendrycks & Gimpel, 2016; Elfwing et al., 2018; Clevert et al., 2016; Xie et al., 2020). In our assessment of GELU's efficacy for heterogeneous FL, we observe a similar conclusion: as shown in Table 1, replacing ReLU with GELU can lead to a significant accuracy improvement of 4.52% in COVID-FL, from 72.92% to 77.44%.

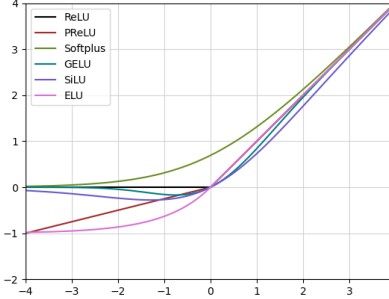

Figure 2: Plots of various activation function curves.

Table 1: A study on the effect of various activation functions.

| Activation | Central | FL | Mean Act |
|------------|---------|-------|----------|
| ReLU | 95.59 | 72.92 | 0.28 |
| LReLU | 95.54 | 73.41 | 0.26 |
| PReLU | 95.41 | 74.24 | 0.20 |
| SoftPlus | 95.28 | 73.86 | 0.70 |
| GELU | 95.82 | 77.44 | 0.07 |
| SiLU | 95.81 | 79.52 | 0.04 |
| ELU | 95.59 | 78.25 | -0.07 |

Building upon the insights from (Xie et al., 2020), we next explore the generalization of this improvement to smooth activation functions, which are defined as being $\mathcal{C}^1$ smooth. Specifically, we assess five activation functions: two that are non-smooth (Parametric Rectified Linear Unit (PReLU) (He et al., 2015) and Leaky ReLU (LReLU)), and three that are smooth (SoftPlus (Dugas et al., 2000), Exponential Linear Unit (ELU) (Clevert et al., 2016), and Sigmoid Linear Unit (SiLU) (Elfwing et al., 2018)). The curves of these activation functions are shown in Figure 2, and their performance on COVID-FL is reported in Table 1. Our results show that smooth activation functions generally outperform their non-smooth counterparts in heterogeneous FL. Notably, both SiLU and ELU achieve an accuracy surpassing 78%, markedly superior to the accuracy of LReLU (73.41%) and PReLU (74.24%). Yet, there is an anomaly in our findings: SoftPlus, when replacing ReLU, fails to show a notable improvement. This suggests that *smoothness alone is not sufficient for achieving strong performance in heterogeneous FL.*

With a closer look at these smooth activation functions, we note a key difference between SoftPlus and the others: while SoftPlus consistently yields positive outputs, the other three (GELU, SiLU, and ELU) can produce negative values for certain input ranges, potentially facilitating outputs that are more centered around zero. To quantitatively characterize this difference, we calculate the mean

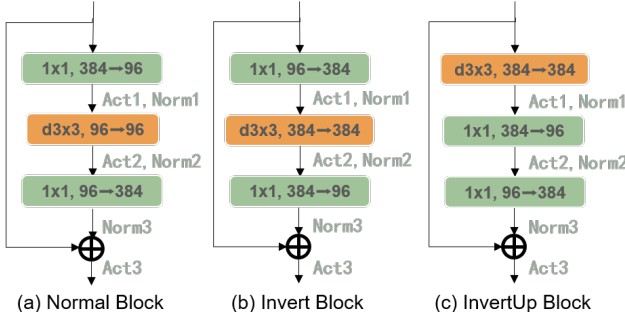

Figure 3: Block architecture details.

activation values of ResNet-M across all layers when presented with COVID-FL data. As noted in the "Mean" column of Table 1, GELU, SiLU, and ELU consistently hold mean activation values close to zero, while other activation functions exhibit a much larger deviations from zero. These empirical results lead us to conclude that utilizing a **smooth and near-zero-centered** activation function is advantageous in heterogeneous FL.

### 3.3 REDUCING ACTIVATION AND NORMALIZATION LAYERS

Transformer blocks, in contrast to traditional CNN blocks, generally incorporate fewer activation and normalization layers (Dosovitskiy et al., 2020; Touvron et al., 2021b; Liu et al., 2021). Prior works show that CNNs can remain stable or even attain higher performance with fewer activation layers (Liu et al., 2022) or without normalization layers (Brock et al., 2021b). In this section, we delve into this design choice in the context of heterogeneous FL.

Drawing inspiration from ConvNeXt, we evaluate three different block instantiations: 1) *Normal Block*, which serves as the basic building block of our ResNet-M baseline; 2) *Invert Block*, originally proposed in (Sandler et al., 2018), which has a hidden dimension that is four times the size of the input dimension; and 3) *InvertUp Block*, which modifies Invert Block by repositioning the depthwise convolution to the top of the block. The detailed block configurations is illustrated in Figure 3. Notably, regardless of the specific block configuration employed, we always ensure similar FLOPs across models by adjusting the number of blocks.

**Reducing activation layers.** We begin our experiments by aggressively reducing the number of activation layers. Specifically, we retain only one activation layer in each block to sustain non-linearity. As highlighted in Table 2a, all three block designs showcase at least one configuration that delivers substantial performance gains in heterogeneous FL. For example, the best configurations yield an improvement of 6.89% for Normal Block (from 77.44% to 84.33%), 3.77% for Invert Block (from 80.35% to 84.12%), and 1.48% for InvertUp Block (from 80.96% to 82.44%). More intriguingly, a simple rule-of-thumb design principle emerges for these best configurations: the activation function is most effective when placed subsequent to the channel-expanding convolution layer, whose output channel dimension is larger than its input dimension.

**Reducing normalization layers.** Building upon our experiments above with only one best-positioned activation layer, we further investigate the impact of aggressively removing normalization layers. Similarly, we hereby are interested in keeping only one normalization layer in each block. As presented in Table 2, we observe that the effects of removing normalization layers are highly block-dependent in heterogeneous FL: it consistently hurts Normal Block, leads to improvements for the Invert Block (up to +2.07%), and consistently enhances InvertUP block (up to +3.26%).

**Normalization-free setup.** Another interesting direction to explore is by removing all normalization layers from our models. This is motivated by recent studies that demonstrate high-performance visual recognition with normalization-free CNNs (Brock et al., 2021a;b). To achieve this, we train normalization-free networks using the Adaptive Gradient Clipping (AGC) technique, following (Brock et al., 2021b). The results are presented in Table 2. Surprisingly, we observe that these normalization-free variants are able to achieve competitive performance compared to their best counterparts with only one activation layer and one normalization layer, *e.g.*, 82.50% *vs.* 82.33% for Normal Block, 84.63% *vs.* 86.19% for Invert Block, and 85.65% *vs.* 85.70% for InvertUP Block.

Table 2: (a) Analysis of the effect of reducing activation functions. "ActX" refers to the block that only keeps the activation layer after the Xth convolution layer. "All" refers to keeping all activation layers within the block. (b) Analysis of the effect of reducing normalization functions. "NormY" refers to the block that only keeps the normalization layer after the Yth convolution layer. "No Norm" refers to removing all normalization layers within the block.

| Block | Act | Central | FL |
|---|---|---|---|
| Normal | All | 95.82 | 77.44 |
| | Act1 | 95.41 | 79.24 |
| | Act2 | 95.41 | 80.12 |
| | Act3 | 95.89 | 84.33 |
| Invert | All | 95.64 | 80.35 |
| | Act1 | 96.19 | 84.12 |
| | Act2 | 95.24 | 82.06 |
| | Act3 | 95.46 | 78.60 |
| InvertUp | All | 95.76 | 80.96 |
| | Act1 | 95.21 | 76.97 |
| | Act2 | 95.71 | 82.44 |
| | Act3 | 95.56 | 77.46 |

(a)

| Block | Act | Norm | Central | FL |
|---|---|---|---|---|
| Normal | Act3 | All | 95.89 | 84.33 |
| | Act3 | Norm1 | 95.84 | 82.06 |
| | Act3 | Norm2 | 95.36 | 82.33 |
| | Act3 | Norm3 | 95.34 | 81.36 |
| | Act3 | No Norm | 95.29 | 82.50 |
| Invert | Act1 | All | 96.19 | 84.12 |
| | Act1 | Norm1 | 95.76 | 82.48 |
| | Act1 | Norm2 | 96.04 | 83.02 |
| | Act1 | Norm3 | 95.59 | 86.19 |
| | Act1 | No Norm | 95.94 | 84.63 |
| InvertUp | Act2 | All | 95.71 | 82.44 |
| | Act2 | Norm1 | 95.64 | 82.45 |
| | Act2 | Norm2 | 95.64 | 83.46 |
| | Act2 | Norm3 | 95.71 | 85.70 |
| | Act2 | No Norm | 95.74 | 85.65 |

(b)

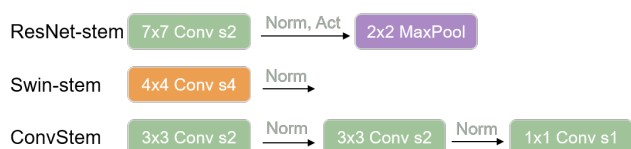

Figure 4: Illustration of different stem setups, including ResNet-stem, Swin-stem and ConvStem. 's' denotes the stride of convolution.

Additionally, a normalization-free setup offers practical advantages such as faster training speed (Singh & Shrivastava, 2019) and reduced GPU memory overhead (Bulo et al., 2018). In our context, compared to the vanilla block instantiations, removing normalization layers leads to a 28.5% acceleration in training, a 38.8% cut in GPU memory, and conveniently bypasses the need to determine the best strategy for reducing normalization layers.

Moreover, our proposed normalization-free approach is particularly noteworthy for its superior performance compared to FedBN (Li et al., 2021) (see supplementary material), a widely acknowledged normalization technique in FL. While FedBN decentralizes normalization across multiple clients, ours outperforms it by completely eliminating normalization layers. These findings invite a reevaluation of "already extensively discussed" normalization layers in heterogeneous FL.

## 3.4 STEM LAYER

CNNs and Transformers adopt different pipelines to process input data, which is known as the *stem*. Typically, CNNs employ a stack of convolutions to downsample images into desired-sized feature maps, while Transformers use patchify layers to directly divide images into a set of tokens. To better understand the impact of the stem layer in heterogeneous FL, we comparatively study diverse stem designs, including the default ResNet-stem, Swin-stem, and ConvStem inspired by (Xiao et al., 2021). A visualization of these stem designs is provided in Figure 4, and the empirical results are reported in Table 3. We note that 1) both Swin-stem and ConvStem outperform the vanilla ResNet-stem baseline, and 2) ConvStem attains the best performance. Next, we probe potential enhancements to ResNet-stem and Swin-stem by leveraging the "advanced" designs in ConvStem.

**Overlapping convolution.** We first investigate the performance gap between Swin-stem and ConvStem. We posit that the crux of this gap might be attributed to the variation in patch overlapping. Specifically, Swin-stem employs a convolution layer with a stride of 4 and a kernel size of 4, thereby extracting features from non-overlapping patches; while ConvStem resorts to overlapping convolu-

Table 3: Analysis of the effect of various stem layers. "(Kernel Size 5)" denotes using a kernel size of 5 in convolution. "(No MaxPool)" denotes removing the max-pooling layer and increasing the stride of the first convolution layer accordingly.

| Stem | Central | FL |
|---|---|---|
| ResNet-stem | 95.82 | 77.44 |
| Swin-stem | 95.61 | 79.44 |
| ConvStem | 95.26 | **83.01** |
| Swin-stem (Kernel Size 5) | 95.64 | 82.76 |
| ResNet-stem (No MaxPool) | 95.76 | 82.59 |

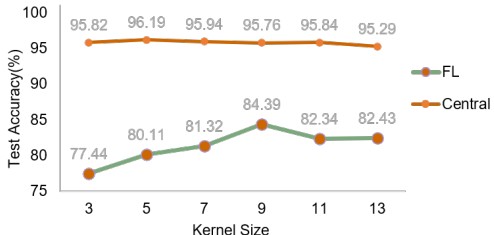

Figure 5: The study on the effect of different kernel sizes.

tions, which inherently bring in adjacent gradient consistency and spatial smoothness (Graham et al., 2021). To validate our hypothesis, we modify the Swin-stem by increasing the kernel size to 5 while retaining a stride of 4. This seemingly modest alteration yielded a marked performance enhancement of +3.32% (from 79.44% to 82.76%), confirming the pivotal role of overlapping convolutions within stem layers in heterogeneous FL.

**Convlutions-only downsampling.** ResNet-stem, despite its employment of a 7×7 convolution layer with a stride of 2 — thereby extracting features from overlapped patches — remarkably lags behind Swin-stem in performance. A noteworthy distinction lies in the ResNet-stem's integration of an additional max-pooling layer to facilitate part of its downsampling; while both Swin-stem and ConvStem exclusively rely on convolution layers for this purpose. To understand the role of the max-pooling layer within ResNet-stem, we remove it and adjust the stride of the initial convolution layer from 2 to 4. As shown in Table 3, this modification, dubbed "ResNet-stem (No MaxPool)", registers an impressive 5.15% absolute accuracy improvement over the vanilla ResNet-stem. This observation suggests that employing convolutions alone (hence no pooling layers) for downsampling is important in heterogeneous FL.

In summary, our analysis highlights the significance of the stem layer's design in securing model performance in heterogeneous FL. Specifically, two key factors are identified, *i.e.*, the stem layer needs to extract features from overlapping patches and employs convolutions only for downsampling.

### 3.5 KERNEL SIZE

Global self-attention is generally recognized as a critical factor that contributes to the robustness of ViT across diverse data distributions in FL (Qu et al., 2022b). Motivated by this, we explore whether augmenting the receptive field of a CNN– by increasing its kernel size — can enhance this robustness. As depicted in Figure 5, increasing the kernel size directly corresponds to significant accuracy improvements in heterogeneous FL. The largest improvement is achieved with a kernel size of 9, elevating accuracy by 6.95% over the baseline model with a kernel size of 3 (*i.e.*, 84.39% *vs.* 77.44%). It is worth noting, however, that pushing the kernel size beyond 9 ceases to yield further performance enhancements and might, in fact, detract from accuracy.

### 3.6 COMPONENT COMBINATION

We now introduce FedConv, a novel CNN architecture designed to robustly handle heterogeneous clients in FL. Originating from our ResNet-M baseline model, FedConv incorporates five pivotal design elements, including *SiLU activation function*, *retraining only one activation function per block*, *normalization-free setup*, *ConvStem*, and *a large kernel size of 9*. By building upon three distinct instantiations of CNN blocks (as illustrated in Figure 3), we term the resulting models as FedConv-Normal, FedConv-Invert, and FedConv-InvertUp.

Our empirical results demonstrate that these seemingly simple architectural designs collectively lead to a significant performance improvement in heterogeneous FL. As shown in Table 4, FedConv models achieve the best performance, surpassing strong competitors such as ViT, Swin-Transformer, and ConvNeXt. The standout performer, FedConv-InvertUp, records the highest accuracy of 92.21%, outperforming the prior art, ConvNeXt, by 2.64%. These outcomes compellingly contest the assertions in (Qu et al., 2022b), highlighting that a pure CNN architecture can be a competitive alternative to ViT in heterogeneous FL scenarios.

Table 4: Performance comparison on COVID-FL. By incorporating archiectural elements such as SiLU activation function, retaining only one activation function, the normalization-free setup, ConvStem, and a large kernel size of 9, our FedConv models consistently outperform other advanced solutions in heterogeneous FL.

| Model | FLOPs | Central | FL |
|---|---|---|---|
| ResNet50 | 4.1G | 95.66 | 73.61 |
| ResNet-M | 4.6G | 95.82 | 77.44 |
| Swin-Tiny | 4.5G | 95.74 | 88.38 |
| ViT-Small | 4.6G | 95.86 | 84.89 |
| ConvNeXt-Tiny | 4.5G | 96.01 | 89.57 |
| FedConv-Normal | 4.6G | 95.84 | 90.61 |
| FedConv-Invert | 4.6G | 96.19 | 91.68 |
| FedConv-InvertUp | 4.6G | 96.04 | 92.21 |

Table 5: Performance comparison on CIFAR-10 and iNaturalist. Our FedConv model consistently outperforms other models. Notably, as data heterogeneity increases, FedConv's strong generalization becomes more evident.

| Model | CIFAR-10 | | | | iNaturalist |
|---|---|---|---|---|---|
| | Central | Split-1 | Split-2 | Split-3 | FL |
| ResNet50 | 97.47 | 96.69 | 95.56 | 87.43 | 12.61 |
| Swin-Tiny | 98.31 | 98.36 | 97.83 | 95.22 | 24.57 |
| ViT-Small | 97.99 | 98.24 | 97.84 | 95.64 | 40.30 |
| ConvNeXt-Tiny | 98.31 | 98.20 | 97.67 | 95.85 | 22.53 |
| FedConv-InvertUp | 98.42 | 98.11 | 97.74 | 96.26 | 54.19 |

# 4 Generalization and Practical Implications

In this section, we assess the generalization ability and communication costs of FedConv, both of which are critical metrics for real-world deployments. To facilitate our analysis, we choose FedConv-InvertUp, our top-performing variant, to serve as the default model for our ablation study.

## 4.1 Generalizing to other Datasets

To assess the generalizability of our model, we evaluate its performance on two additional heterogeneous FL datasets: CIFAR-10 (Krizhevsky et al., 2009) and iNaturalist (Van Horn et al., 2018).

**CIFAR-10.** Following (Qu et al., 2022b), we use the original test set as our validation set, and the training set is divided into five parts, with each part representing one client. Leveraging the mean Kolmogorov-Smirnov (KS) statistic to measure distribution variations between pairs of clients, we create three partitions, each representing different levels of label distribution skewness: split-1 (KS=0, representing an IID set), split-2 (KS=0.49), and split-3 (KS=0.57).

**iNaturalist.** iNaturalist is a large-scale fine-grained visual classification dataset, containing natural images taken by citizen scientists (Hsu et al., 2020). For our analysis, we use a federated version, iNature, sourced from FedScale (Lai et al., 2021). This version includes 193K images from 2295 clients, of which 1901 are in the training set and the remaining 394 are in the validation set.

Table 5 reports the performance on CIFAR-10 and iNaturalist datasets. As data heterogeneity increases from split1 to split3 on CIFAR-10, while FedConv only experiences a modest accuracy drop of 1.85%, other models drop the accuracy by at least 2.35%. On iNaturalist, FedConv impressively achieves an accuracy of 54.19%, surpassing the runner-up, ViT-Small, by more than 10%. These results confirm the strong generalization ability of FedConv in highly heterogeneous FL settings.

## 4.2 Generalizing to other FL Methods

We next evaluate our model with different FL methods, namely FedProx (Li et al., 2020c), FedAVG-Share (Zhao et al., 2018), and FedYogi (Reddi et al., 2020). FedProx introduces a proximal term to estimate and restrict the impact of the local model on the global model; FedAVG-Share utilizes a

globally shared dataset to collect data from each client for local model updating; FedYogi incorporates the adaptive optimization technique Yogi (Zaheer et al., 2018) into the FL context.

The results, as reported in Table 6, consistently highlight the superior performance of our FedConv model across these diverse FL methods. This observation underscores FedConv's potential to enhance a wide range of heterogeneous FL methods, enabling seamless integration and suggesting its promise for further performance improvements.

Table 6: Performance comparison with different FL methods on COVID-FL. 'Share' denotes 'FedAVG-Share'. We note our FedConv consistently shows superior performance.

| Model | FedProx | Share | FedYogi |
|---|---|---|---|
| ResNet50 | 72.92 | 92.43 | 66.01 |
| ViT-Small | 87.07 | 93.89 | 87.69 |
| Swin-Tiny | 87.74 | 94.02 | 91.86 |
| ConvNeXt-Tiny | 89.35 | 95.11 | 92.46 |
| FedConv-InvertUp | 92.11 | 95.23 | 93.10 |

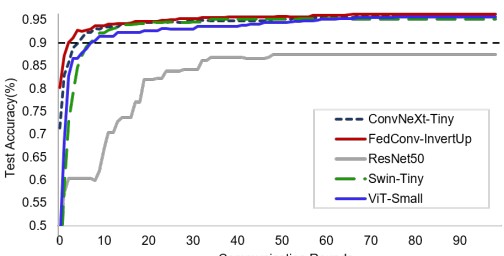

Figure 6: Test accuracy versus communication rounds conducted on the split-3 of CIFAR-10. The black dashed line is the target test accuracy. Our model shows the fastest convergence speed.

### 4.3 COMMUNICATION COST

In FL, communication can be a major bottleneck due to the inherent complications of coordinating numerous devices. The process of client communication is often more time-consuming than local model updates, thereby emerging as a significant challenge in FL (Van Berkel, 2009). The total number of communication rounds and the size of the messages transmitted during each round are key factors in determining the communication efficiency (Li et al., 2020b). To comprehensively evaluate these aspects, we follow the methodology proposed in (Qu et al., 2022b). Specifically, we record the number of communication rounds required for different models to achieve a preset accuracy threshold. Additionally, we use Transmitted Message Size (TMS), which is calculated by multiplying the number of model parameters with the associated communication rounds, to quantify communication costs.

As shown in Figure 6, our FedConv-InvertUp achieves the fastest convergence speed among all models. In CIFAR-10 split3, where high data heterogeneity exists, FedConv-InvertUp only needs 4 communication rounds to achieve the target accuracy of 90%, while ConvNeXt necessitates 7 rounds. This efficiency also translates to a marked reduction in TMS in FL, as reported in Table 7. In contrast, ResNet struggles to converge to the 90% accuracy threshold in the CIFAR-10 split3 setting. These results demonstrate the effectiveness of our proposed FedConv architecture in reducing communication costs and improving the overall FL performance.

Table 7: Comparison based on TMS. TMS is calculated by multiplying the number of model parameters with the communication rounds needed to attain the target accuracy. We note our FedConv requires the lowest TMS to reach the target accuracy.

| Model | ResNet50 | Swin-Tiny | ViT-Small | ConvNext-Tiny | FedConv-InvertUp |
|---|---|---|---|---|---|
| TMS | $\infty$ | 27.5M×10 | 21.7M×11 | 27.8M×8 | 25.6M×5 |

### 5 CONCLUSION

In this paper, we conduct a comprehensive investigation of several architectural elements in ViT, showing those elements that, when integrated into CNNs, substantially enhance their performance in heterogeneous FL. Moreover, by combining these architectural modifications, we succeed in building a pure CNN architecture that can consistently match or even outperform ViT in a range of heterogeneous FL settings. We hope that our proposed FedConv can serve as a strong baseline, catalyzing further innovations in FL research.

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
