# FedConv: Enhancing Convolutional Neural Networks for Handling Data Heterogeneity in Federated Learning

## Supplementary Material

## A   Appendix

### A.1   Analysis of FedBN

We have also experimented with FedBN(Li et al., 2021), a popular algorithm that also tackles data heterogeneity from the perspective of model architecture. Specifically, it proposes to not average BN layers in FedAVG. We apply this method on the regular ResNet-50, and a ConvNeXt-Tiny model with its normalization layer changed from LN-C to BN. As shown in Table 1, in split 1 and 2, where data heterogeneity is not so severe, FedBN achieves close performance compared to FedAVG. However, in a more extreme heterogeneous scenario like CIFAR-10 split3, the performance of both ResNet and ConvNeXt-BN trained by FedBN drops sharply. Specifically, from split1 to split 3, ResNet and ConvNeXt-BN shows a drop of 14.60% (96.42% to 81.82%), and 24.12% (97.99% to 73.87%), respectively. By contrast, the original ConvNeXt that chooses LN-C as its normalization layer, shows only a performance drop of 2.35% (98.20% to 95.85%). Also, our proposed FedConv achieves the best accuracy of 96.26% in split3, demonstrating the effectiveness of the normalization-free design.

Table 1: Performance comparison on CIFAR-10 dataset. 'ConvNeXt-BN' denotes ConvNeXt with LN-C changed to BN. '$\pm$' indicates the range of accuracy between clients.

| Method | Model | Split1 | Split2 | Split3 |
|--------|-------|--------|--------|--------|
| FedBN | ResNet50 | 96.42$\pm$0.18 | 93.15$\pm$1.35 | 81.82$\pm$1.52 |
| | ConvNeXt-BN | 97.99$\pm$0.05 | 95.76$\pm$0.82 | 73.87$\pm$12.57 |
| FedAVG | ResNet50 | 96.69$\pm$0.00 | 95.56$\pm$0.00 | 87.43 $\pm$0.00 |
| | ConvNeXt | 98.20$\pm$0.00 | 97.67$\pm$0.00 | 95.85$\pm$0.00 |
| | FedConv-InvertUp | 98.11$\pm$0.00 | 97.74$\pm$0.00 | 96.26$\pm$0.00 |

### A.2   Implementation Details

#### A.2.1   Pre-training Recipe

Following Liu et al. (2022), we pre-train our model for 300 epochs. The learning rate is set to 4e-3 with linear warmup for 20 epochs and cosine decay schedule in subsequent epochs. AdamW (Loshchilov & Hutter, 2019) is adopted with weight decay set to 0.05. For data augmentation, we adopt Mixup (Zhang et al., 2018), CutMix (Yun et al., 2019), RandAugment (Cubuk et al., 2020) and Random Erasing (Zhong et al., 2020). For regularization, we adopt Stochastic Depth(Huang et al., 2016) and Label Smoothing (Szegedy et al., 2016). Layer Scale (Touvron et al., 2021) of initial value 1e-6 is applied. All layer weights are initialized using truncated normal distribution.

#### A.2.2   Fine-tuning Recipe

**CIFAR-10.** Following Qu et al. (2022), for CNN structures and ViT, a SGD optimizer without weight decay is adopted. For Swin-Transformer, a AdamW with a weight decay of 0.05 is adopted. The learning rate is set to 0.03 for CNNs and ViT, and 3.125e-5 for Swin-Transformer. We use 5-epoch linear warmup and cosine learning rate decay. Stochastic Depth rate is applied with value

set to 0.1, and gradient clipping is set to 1 for all models. For FL settings, we train models for 100 communication rounds, and we choose all 5 clients in each round.

**iNaturalist.** We follow the training settings in CIFAR-10. For FL settings, we train models for 2000 communication rounds, and choose 25 clients in each round.

### A.2.3 FL METHODS

In FedProx, $\mu$ is set to 5e-5 for Swin, and 5e-4 for other models. We share 5% images from each client in FedAVG-Share. In FedYogi, we set $\beta_1$, $\beta_2$ to 0.9 and 0.99 following Reddi et al. (2020), client learning rate $\eta$ to 0.01, and adaptivity $\tau$ to 5e-2 for ResNet50, 4e-3 for other models.