# OpenReview forum: "FedConv: Enhancing Convolutional Neural Networks for Handling Data Heterogeneity in Federated Learning"
_ICLR.cc/2024/Conference — ICLR 2024 Conference Withdrawn Submission_

### Official Review · Reviewer_4TD6 · 2023-10-27

**Soundness:** 3 good
**Presentation:** 3 good
**Contribution:** 3 good
**Rating:** 6
**Confidence:** 4

**Summary:**

This paper presents an empirical investigation into strategies for mitigating heterogeneity within the context of FL using CNNs. The main focus is directed towards elucidating the different components in CNNs such as activation functions, block architecture design, normalization techniques, and stem layer configurations. The paper demonstrates the potential for carefully designed CNN architectures to yield better performance when compared to ViT-based models in heterogeneous FL. Moreover, their approach also generalizes well to various FL methods and demonstrates notable advantages in terms of reducing communication overheads.

**Strengths:**

**Quality & Clarity**: The paper is well-organized, with each subsection providing a clear introduction to the specific component under examination. It follows with tables and figures to effectively support their findings.

**Originality**: The exploration of various components in CNNs to boost performance in heterogeneous FL is a novel concept. It offers insights into the architectural considerations for heterogeneous FL. Additionally, the comparative analysis between ViTs and CNNs introduces intriguing perspectives on the pros and cons of distinct architectural approaches.

**Significance**: The advanced performance achieved by carefully designed CNNs compared to ViTs in the context of heterogeneous FL serves as a compelling reminder of the significance of CNNs, even in an era where ViTs have gained dominance in the field of deep learning.

**Weaknesses:**

1. This paper primarily relies on empirical insights and lacks a comprehensive theoretical analysis. It is difficult to understand the underlying principles that drive the efficacy of this method. (see Q1)

2. The main content of this paper is centered around the examination of large models like ConvNeXt and Swin ViT. The potential applicability of the proposed methods to models of varying sizes remains uncertain, which holds significant importance given the large body of FL for training models on edge devices, such as mobile phones.

3. Some of the implementation details are missing. (see Q2, Q3)

**Questions:**

1. The designs discussed in the paper lead to improvement in heterogeneous FL, but no difference for the central model. What is the possible reason that makes such designs so effective in FL? For example, in Table 3, why increasing the kernel size can effectively increase the accuracy of Swin-stem in FL setting, while there is no difference for the central model?

2. How many classes does the COVID-FL dataset have? How are the samples distributed among different clients?

3. (a) In section 3.4, how does "stem" work? Are the rest of the models the same except for the stem layer?
(b) For stem layers, when changing the kernel size for Swin-stem, or replacing the max-pooling with convolution layers in ResNet-stem, do the FLOPs change? If it changes, is it still considered a fair comparison with changed FLOPs?

4. What is the result of the central model for iNaturalist? Since the gap between Central and FL is small for both COVID-FL and CIFAR-10, I wonder if it is the same for iNaturalist dataset.

---

### Official Review · Reviewer_nqDj · 2023-10-27

**Soundness:** 2 fair
**Presentation:** 2 fair
**Contribution:** 2 fair
**Rating:** 3
**Confidence:** 4

**Summary:**

The paper studies neural architectural components to create a model architecture that is robust against data heterogeneity in FL. The main motivation is derived from ViTs in a sense that as per Qu et al. (2022b) ViTs perform better when data is heterogeneous. The paper is more or less empirical in nature and on a high-level seems like a hyper-parameter optimization being done in federated setting when data is nonIID.

**Strengths:**

The proposed architectural blocks are simple and easy to understand. The paper takes a different approach to more recent studies to study how different blocks behave in federated setting when the data is heterogeneous.

**Weaknesses:**

The self-attention free CNN is also very similar to ConvMixer and this is not a new observation. Can you highlight how FedConv is any different and how the involved components are discovered/developed? The motivation is largely lacking and it looks like these component are a result of running a large-scale hyperparameter optimization?

Number of clients, i.e., 12, are too low to make any strong claims about data heterogeneity. Why not systematically vary the number of clients from 10 to 200?

No comparison with existing approaches. It looks like this paper is a first attempt at studying data heterogeneity in federated setting.

**Questions:**

Why are no numbers mentioned around the data being heterogeneous in the second-to-last paragraph to better contextualize the accuracies across various datasets? Either mention both, so the reader can better understand the performance early on, or leave them entirely for the results section.

What happens if one does not pretrain the model? To me, this looks like a central aspect of the method as well. If pretraining is necessary, why not compare it against other pretrained models like the CLIP or SigLIP vision encoder?

What is "diverse input"? Diverse in what sense?, paragraph 3 of section 1. How do you control for diversity?

Why is there no comparison with existing methods? This is the most important in my opinion, and it is very hard to judge the usefulness of the proposed model without an extensive comparison with other approaches.

How would this model compare against training ResNet50/ViT (from CLIP) with FedProx or similar method?

What is the main motivation for only using 3 splits of CIFAR-10? Why is there no evaluation on CIFAR-100 with varying \alpha, for example, setting \alpha to 100, 1, and 0.01 to control from nonIIDness?

What is the impact on performance if one varies $C$, the fraction of clients that can participate in each round?

From Table 5, the difference in results (for CIFAR-10) with other model architectures is very small. The improvement is not even significant. Why is ConvNeXt-Tiny performing worse on iNaturalist when its performance is on par with FedCov for CIFAR-10? Are the number of parameters in ConvNeXt-Tiny and FedConv-InvertUp the same? Are both models trained with the same configuration?

I also want to understand the reasoning behind plotting test set accuracy vs. communication round. In a centralized setting, would it be acceptable to plot test set accuracy vs. the number of epochs? If not, why is it considered ok in a federated setting? Additionally, based on this figure, it seems that one can simply stop training after 15+ rounds and still achieve better performance than the target accuracy. Are these kind of plots used to select number of rounds and other hyperparameters?

Nit: The claim that there are "studies" showing ViTs are more robust against data heterogeneity in FL is too strong, as there is actually only one study by Qu et al. (2022b) that supports this. It is suggested to rephrase the text.

---

### Official Review · Reviewer_vWDW · 2023-10-28

**Soundness:** 3 good
**Presentation:** 3 good
**Contribution:** 2 fair
**Rating:** 3
**Confidence:** 5

**Summary:**

This paper comprehensively evaluated the effect of different components of DNNs on heterogeneous data federated learning. Based on their empirical, experimental results, the paper further proposes the neural architecture design principle guidelines for FL. Leveraging the proposed guideline, pure CNNs can achieve competitive performance with ViTs in data heterogeneity FL.

**Strengths:**

- Experiments with different neural network components and investigate their importance for heterogeneous data FL
- Propose specialized neural architecture design strategy for heterogeneous data FL.
- Based on the proposed architecture design principle, this paper further designed a pure CNN that achieves competitive performance as ViT

**Weaknesses:**

**Proposed principle – removing all normalization layers and normalization-free setup**:

The paper conducts an experiment to show the effectiveness of removing all normalization layers only on the neural architecture view and lacks consideration for other factors.

It's important to note that the aim of normalization technique is to create faster and more stable training, while improving generalization (i.e generalization performance of the network on unseen data.)

The normalization layer is highly "model size and dataset" aware; for instance, the larger the dataset, the superior results with the normalization you will get. In your experiments, for instance, the normalization technique in ResNet is targeted to a larger dataset i.e ImageNet. So instead of simply keeping and removing normalization, you should also consider the dataset and mode size. Since FL is a complex system, some clients may run large-size models, and simple removal of normalization from a large model might easily go to over-fittings.

In short, more comprehensive factors need to be discussed.


**Proposed principle – Reducing activation layers**:

The effect of reducing activation is similar to adjusting the size of Deep Neural Networks ( reducing the `depth` of DNNs and increasing the `width` of the DNN).

For instance, let's consider 2 hidden layers in DNNs: layer 1 with weights W1, and layer 2 with weights W2. The calcuation with activation $\sigma$: $$Y =\sigma(W_2 \sigma(W_1 X))$$

if we remove the activation between layer 1 and 2, the computation between layer 1 and 2 will become:
$$Y = \sigma(W_2 W_1 X)$$
According to the associative property of matrix multiplication, we can have $Y = \sigma((W_2 W_1) X)$,  and let  $W_3 =W_2 W_1 $, then we have $Y = \sigma(W_3 X)$. Hence, it's similar to replacing layer 1 and layer 2 with a new layer with learnable weights $W_3$

Again, in this paper's case, the effect of removing activation layers is essentially we modify the DNNs' size (the depth and width), and the size of the DNNs is a complex hyper-parameter tuning and AutoML topic, it depends on many factors and no universal answers.

**Proposed principle – Kernel Size**:
The paper [1] also has a similar conclusion, they use large kernel design 31*31, but not in FL settings, and achieve competitive performance with ViT.  However, selecting the right kernel size raises trade-offs among model size, computational complexity, memory usage, etc. It's not always universal for the larger kernel size the better, it depends on specific questions and tasks.

**The interpretability of neural architecture in FL shouldn't separate from existing research**:

It's very hard to separate the interpretability of the neural architecture in FL from existing research, again, FL systems are complex and dynamic, and it's challenging to find a universal guideline for all cases. Even the authors in this paper tried to limit their domain to heterogeneous data (Non-IID) FL, the experiment settings are primarily focused on neural architecture view. and lack the ablations on different data-heterogeneous settings. I list some of the references for investigating the effect and interpretability of neural architecture components for general model optimization guidelines. [1, 2, 3, 4]


**Unfair design of experiments**:

Since FL challenges traditional centralized training with its heterogeneous data, simply active and removing some components of DNNs is unfair for investigating the effect in FL. Different non-IID settings need to be discussed. For instance, the normalization designed for overfittings and generalizability, and the overfitting and generalizability should also be discussed, even in an FL setup.


**Reference**

[1]Ding, Xiaohan, et al. "Scaling up your kernels to 31x31: Revisiting large kernel design in cnns." Proceedings of the IEEE/CVF conference on computer vision and pattern recognition. 2022.

[2] Ş. Öztürk, U. Özkaya, B. Akdemir and L. Seyfi, "Convolution Kernel Size Effect on Convolutional Neural Network in Histopathological Image Processing Applications," 2018 International Symposium on Fundamentals of Electrical Engineering (ISFEE), Bucharest, Romania, 2018, pp. 1-5, doi: 10.1109/ISFEE.2018.8742484.

[3] Yu, Jiahui, and Konstantinos Spiliopoulos. "Normalization effects on deep neural networks." arXiv preprint arXiv:2209.01018 (2022).

[4]Liu, Sheng, et al. "Convolutional normalization: Improving deep convolutional network robustness and training." Advances in neural information processing systems 34 (2021): 28919-28928.

**Questions:**

Please address my concern in the Weakness section, besides that:

I like the idea of this paper since model model compression and architecture design are growing important to edge-FL scenarios.
However, addressing the interoperability issue of neural networks shouldn't be separate from the existing research. Readers are more expected to see the heterogeneous data (Non-IID)'s effect for mode architecture components.  Hence, experiments with various Non-IID levels of settings are essential to improve the fairness of the experiments.

---

### Official Review · Reviewer_Ggo2 · 2023-10-29

**Soundness:** 3 good
**Presentation:** 3 good
**Contribution:** 3 good
**Rating:** 6
**Confidence:** 4

**Summary:**

Based on the various performance deteriorations of CNN and ViT in FL settings, this paper analyzes the architecture of ViTs and uncovers the designs that can alleviate the data heterogeneity problem. Furthermore, these designs are integrated into the traditional CNN architecture to build a self-attention-free CNN architecture (FedConv). Experiments show the performance improvement achieved by FedConv.

**Strengths:**

1. This paper poses an interesting and novel research direction. It is worthwhile to investigate how the model architecture designs can affect the robustness to the heterogeneity problem.

2. Significant improvement in test performance is achieved after adjusting the model architecture according to the conclusions, demonstrating the effectiveness of the empirical observation.

3. This paper sheds light on the model architecture configuration for existing CNN architectures in FL.

**Weaknesses:**

1. How the authors are motivated to explore and lead to the findings can be further clearly explained. For example, there are many model configurations for a specific CNN model, why activation function and normalization layers are selected as key components to explore can be further justified.

2. The number of parameters in CNN and ViT differs a lot, which can lead to different representation abilities of CNN and ViT. This aspect should be taken into consideration when comparing these two architectures.

**Questions:**

Is there any chance to further extend the existing method to a model heterogeneity scenario? For example, different clients own different CNN architectures while training a model in a federated manner. Can this challenging scenario still benefit from the findings of this paper?